# Varied Effects of COVID-19 Chemosensory Loss and Distortion on Appetite: Implications for Understanding Motives for Eating and Drinking

**DOI:** 10.3390/foods11040607

**Published:** 2022-02-20

**Authors:** Lydia Turner, Peter J. Rogers

**Affiliations:** Nutrition and Behaviour Unit, School of Psychological Science, University of Bristol, Bristol BS8 1TU, UK; lydiaturner600@gmail.com

**Keywords:** smell, taste, chemesthesis, appetite, COVID-19, anosmia, ageusia, parosmia, alcohol, flavour

## Abstract

A common symptom of COVID-19 is altered smell and taste. This qualitative study sought to further characterise this altered chemosensory perception and its effects on appetite for food and drink. Eighteen women and two men who had experienced chemosensory loss associated with COVID-19 participated in semi-structured interviews. Thematic analysis of the interview transcripts revealed five major themes. These confirmed that all participants had experienced an altered sense of smell (anosmia, and less frequently parosmia and phantosmia) of variable duration. Loss of taste (ability to detect sweetness, saltiness, etc.) was less common. Participants experienced decreased, no change or increased appetite, with six participants reporting weight loss. Consistent with evidence linking diminished appetite with inflammation, for two participants, decreased appetite preceded anosmia onset. Anosmia reduced enjoyment of food and drink. Compensatory strategies included choosing salty, sweet and ‘spicy’ foods, and increased attention to food texture, and there was evidence that the postingestive rewarding effects of food intake were also important for maintaining appetite. Some participants mentioned increased alcohol intake, in part facilitated by reduced intensity of disliked flavours of alcoholic drinks. The narratives also underlined the value placed on the sociability and structuring of time that daily meals provide. This research adds to the record and analysis of lived experiences of altered chemosensory perception resulting from SARS-CoV-2 infection, and it contributes insights concerning the role of smell and flavour in motivating and rewarding food ingestion.

## 1. Introduction

A common symptom of COVID-19 is altered smell and taste, including loss and distortion of smell and taste [1,2,3,4,5,6]. Given the importance of taste and smell in motivating and guiding eating behaviour [7,8,9,10,11], these symptoms can be expected to have significant effects on food choice and food intake, and in turn adversely affect nutritional status if changes in taste and smell are long lasting. The present study sought to add to the record of lived experiences of participants for whom altered taste and smell was a confirmed symptom of COVID-19, with a particular focus on characterising the nature of the changes in taste and smell perception, and on documenting and interpreting the nature and extent of associated changes in appetite and food choice.

To set the scene for this research, the next three paragraphs describe relevant aspects of chemosensory (taste and smell) perception and its role in motivating and guiding eating behaviour. The word ‘taste’, as used in English, can refer to the flavour of food in the mouth. The experience of flavour during the mastication of food is dependent on the integration of predominantly two sources of chemosensory information, originating from olfactory receptors, detecting volatile compounds in the retronasal airflow, and receptors on the tongue, soft palate and throat detecting compounds giving rise to sweet, salty, sour, bitter and umami (savoury) sensations. Additionally, contributing to flavour are compounds that have been summarised as ‘chemical irritants’, which are detected by oral chemoreceptors, primarily giving rise to the sensation of temperature and pain (responsible, for example, for the cooling sensation of menthol, the burn of chili and the tingling of carbonation) [2,10,12,13]. In this respect, flavour can be thought of as a ‘gestalt’, although we can also attend to and learn to identify individual features of the whole object (the sweetness and notes of cherry and tannin in a wine, for example). Food texture is a further important attribute of food sensed in the mouth, which, via its effect on mastication (oral processing), influences flavour release and rate of eating [14,15,16].

The confusion and overlap of taste and smell is well known and has been discussed in relation to sources of flavour (‘taste’) being co-located with the presence of food felt in the mouth, in contrast to smells originating from the external environment (i.e., orthonasally) [10,17]. Therefore, for the avoidance of doubt in summarising, describing and discussing the narratives generated in present study, we will avoid the ambiguity of the word ‘taste’ by referring specifically to olfaction (sense of smell), gustation (sensation of sweetness, saltiness, etc.) and chemesthesis (sensation of the heat of chili, etc.). Furthermore, where necessary, we will differentiate between orthonasal and retronasal olfaction. These considerations are important because, as we discuss later, confusion between taste and smell potentially causes effects on gustation to be overestimated, even, for example, in studies in which participants are asked to report specifically about altered sensation of sweetness, saltiness, etc., e.g., in [4,5,18]. Altered sense of smell is more straightforwardly diagnosed by reference to orthonasal olfaction, that is, to changes in the perception of smells, such perfume, smoke and baking bread, emanating from objects in the external environment.

In addition to their identity, we experience food smells, tastes and textures hedonically, that is, in relation to how pleasant we find them and how much pleasure we derive from them. This is also expressed as how much we like those attributes. Food liking is greatly affected by exposure, and at least part of that is underpinned by association of the taste of food or drink with their postingestive aftereffects [11,19,20,21,22]. This ‘flavour-postingestive-consequences’ learning changes the hedonic value of flavours but, at least largely, not their identity (for example, through repeated exposure, the bitterness of coffee becomes liked, not less bitter). Additionally, in humans there is also strong evidence for an inborn liking of sweetness, saltiness and umami, and a dislike of sourness and bitterness [23], which can, to some extent, be modified by experience [19,20,21,22]. There is unresolved debate about the function of these inborn taste biases [24]; nonetheless, it is clear, for example, that in nature, sweetness signals the presence of sugars (calories) in a potential foodstuff, even if it does not reliably signal its total nutrient or calorie content [25]. The relevance here is that altered olfaction and gustation, and especially when this occurs suddenly, can be expected to have a profound effect on the pleasure (or ‘enjoyment’) associated with eating and drinking. We return to this theme in more detail in the Discussion section.

Previous studies of altered chemosensory perception associated with COVID-19 have described loss of smell as a common symptom of COVID-19 [1,2,3,4,5,6,7,18]. Complete loss of olfactory perception is termed anosmia, and reduced olfactory perception is termed hyposmia [26]. Parosmia, the altered perception of a familiar odour, and phantosmia, the perception of a smell in the absence of an odour, are also associated with COVID-19, e.g., [6,18,27]. As described above, there is less certainty about ageusia and hypogeusia, respectively the loss and reduction of gustatory perception, and altered chemesthesis, in relation to COVID-19. The data generated by our study were the transcripts of semi-structured interviews with 20 participants who had experienced chemosensory loss associated with COVID-19. The narratives provided a rich account of the effects of altered chemosensory perception on the participants’ well-being, particularly in respect of their enjoyment of eating, effects on their appetite and how they compensated for this with altered food and drink choices.

## 2. Materials and Methods

### 2.1. Participants

A total of 20 participants (18 female, 2 male), aged 21–68 (mean ± SD: 31.2 ± 14.3) years, were recruited using social-media-based advertisements and snowball sampling. The final sample involved university students and members of the general public. The sample size was based on achieving saturation point in the thematic analysis to ensure detailed and inclusive data.

### 2.2. Ethics

Ethical approval for the study was received from the School of Psychological Science Research Ethics Committee, University of Bristol (ethical approval code: 181120112384). Informed consent was acquired from all participants via email.

### 2.3. Procedure

Participants were interviewed individually via a recorded remote video (Zoom) session. Participants read an information sheet and completed a consent form prior to the interview to confirm that they understood the purpose of the study and were happy to participate. They gave consent for interviews to be recorded on an encrypted password-protected recording device. The interviews were semi structured. Open questions focused on the nature of altered chemosensory perception and related changes to appetite, experiences of eating, eating behaviour and well-being, with prompts to elicit further information as appropriate. The standard questions and prompts used are summarised in Table 1; however, the questions, prompts and probes were flexible, depending on participant responses. Therefore, actual conversations often deviated from the standard interview schedule. The interviews lasted between 9 and 25 min. Participants were debriefed promptly via email. That communication also invited participants to ask any questions or raise any concerns that may have arisen from the interview.

### 2.4. Data Analysis

Recordings for each interview were transcribed verbatim by the researcher. To guarantee anonymity, original recordings were erased post-transcription, and participants were allocated pseudonyms using a random name generator. All other identifying information was changed to protect confidentiality. The transcripts, comprising 31,000 words of text, were analysed using thematic analysis [28], familiarising with the full sample data set by reading and re-reading. An initial coding framework was established pre-analysis to ensure direction in theme consensus. Familiarisation then allowed for the definitive coding frame to be established and applied to the final transcription data set. Thematic analysis continued using an iterative process until no new themes emerged from the data set, terminating data collection at saturation point.

## 3. Results

Following final clarification, familiarisation and the focus of our research, five primary themes emerged from the thematic analysis. These themes were: altered chemosensory perception, appetite (and body weight), altered food choices, drinks, and eating rituals.

The evidence pertaining to the first two of these themes is summarised in Table 2 and is exemplified, together with the evidence for the third, fourth and fifth themes, in the extracts from the interviews presented below. We also collated responses to the question about coping with altered taste and smell perception. 

### 3.1. Theme 1. Altered Chemosensory Perception

#### 3.1.1. Olfaction (Anosmia, Hyposmia, Parosmia and Phantosmia)

The interviews confirmed that altered olfaction was experienced by all participants, which included profound loss of smell (anosmia) for at least one day during their illness, or which, for one participant (Laurissa), comprised primarily parosmia lasting 10 days followed by phantosmia. As summarised in Table 2, the duration of altered olfaction was bimodally distributed, with the majority of observations falling within the range of 1 day (one participant) to 4 weeks. For three participants, anosmia or hyposmia was present for between 4 and 6 months. Median duration of anosmia was 14 days. 

Anosmia was evidenced by, for example, the following descriptions.


*‘…the first time I noticed that my smell was affected I was washing clothes and I always smell the conditioner and I couldn’t smell anything.’ (Emma)*



*‘…like when you have a cold and… you can’t really smell anything, it just progressively got worse, and then I didn’t have any smell at all.’ (Emersyn)*



*‘…my mum sent me these gorgeous freesias that I know smell beautiful. I had to say to my husband do these smell? And he was like, yes they smell lovely.’ (Kallie)*



*‘…because if you don’t have a blocked nose you expect to be able to smell. So when you can’t smell anything, it’s really, really peculiar.’ (Starla)*



*‘I just like went to the kitchen and got like the strongest smelling thing I could think of, like I got like a jar of Marmite and I tried to smell it and I couldn’t like smell a single thing…’ (Ella)*



*‘…the dog, I can’t smell the dog, which is quite nice.’ (Melba)*



*‘…we drive past this landfill site and my mum made a comment about it, and I couldn’t smell it at all.’ (Marie)*



*‘…the loss of smell came around 6 pm, it was literally just like instant, like one minute I could smell and next minute I was like oh god I can’t smell anything.’ (David)*



*‘I remember we were having a bonfire at home… I was standing right in the smoke and I couldn’t smell anything, and it was at that point that I knew I lost my (sense of) smell.’ (Martha)*



*‘…my sense of smell also left but slowly over maybe 48 h, like at six hours I could smell a little bit, and then I held a Vick under my nose and I couldn’t smell it…’ (Norene)*



*‘…it was summer last year, and I couldn’t smell sunscreen. It’s those little things, but I missed the smell of food.’ (Meredith)*



*‘…we have a really strong candle in our kitchen, and everyday I’d get up and smell that to see if it had come back yet, and I’d be like oh I still can’t smell that.’ (Dianne)*


Parosmia was evidenced by, for example, the following descriptions.

*‘…any food that was salty or had a bit of tang to it,* (or) *urine, coffee all smelled like marmite, and so it’s been messing with my mind, and like red wine and everything used to smell like marmite.’ (Emersyn)*


*‘I don’t know what it is about oil, it’s just got a really strong smell to me now… it’s like rancid smell so if anything is really cooked in oil I don’t really like it anymore… also eggs so just like the flavour of them like how I remember has completely changed basically. …smell is distorted so like perfume didn’t smell very nice anymore, so I don’t wear perfume.’ (Angela)*



*‘I had this really strong smell of petrol just like intermittently even like last week everything just smelt really gross.’ (Martha)*


Phantosmia was evidenced by, for example, the following descriptions.


*‘It smelt like something was burning in my nose, so that was all I could smell like a burning smell.’ (Kallie)*


*‘Something very weird that I forgot about happened after I felt like it* (parosmia) *had gone, and this was really odd, is I kept smelling smoke everywhere like cigarette smoke or smoke, and I kept asking everyone if they could smell smoke... So either I just got a phantom smell of smoke …or my smell was more acute.’ (Laurissa)*


*‘I had a really weird phantom smell though where I could smell like bin juice, I smelt it like everyday for like a week and it was like literally in the back of my nose.’ (Angela)*



*‘Like I’d smell random things that weren’t there… I’d be sat at work and I could like smell burning toast or something, or like fish, it was pretty grim…’ (Melba)*


#### 3.1.2. Gustation

Evidence concerning altered gustatory function was more difficult to evaluate than evidence for anosmia. Participants described loss of smell (orthonasal olfaction) and loss of taste. In relation to the latter, there was evidence that the loss of taste was due to loss of retronasal olfaction, with gustation being unaffected, or at least largely unaffected. For example:

*‘I think it was the* (loss of) *flavour more than salty or sweet tastes.’ (Emma)*


*‘…we had a pizza, and I said gosh this tastes really salty.’ (Kallie)*



*‘I couldn’t taste anything other than if things were savoury or sweet…’ (Wendy)*



*‘I could taste the sweetness and the saltiness, but I couldn’t taste the actual flavour.’ (Regana)*



*‘I’d buy salt and vinegar pringles just to like physically kind of make my tongue tingle.’ (Angela)*



*‘I could basically only really taste sourness, saltiness, slightly sweetness.’ (Marie)*



*‘…it all sort of tastes the same, it’s all like kind of simple flavours if that makes sense, like sweet salty, hot or cold and that’s it.’ (David)*


For three participants there was evidence of ageusia.


*‘I couldn’t even taste like salt or sweetness, it was literally like nothing…’ (Starla)*



*‘I couldn’t even taste like sweet or salty, not for at least a month if not longer.’ (Melba)*



*‘I couldn’t taste anything, not even salty or sweet or anything.’ (Kimberley)*



*For one participant there was evidence of a specific loss of bitter taste perception.*



*‘I can only taste if something is salty has sugar in or if something is spicy. And not bitter, bitter stuff doesn’t taste bitter. …in like takeaways and stuff I can taste MSG.’ (Emersyn)*


For one participant there was evidence of a specific loss of sweet taste perception.


*‘Salty and sweet, I wanted to eat salty stuff rather than sweet I don’t know why, I think because I could definitely taste a bit of salt, whereas I couldn’t taste anything else.’ (Martha)*


In some instances, references to complete loss of taste were nonetheless qualified by confirmation that gustatory perception was, at least partly, intact. For example:


*‘Everything, there was just nothing to it. I can’t taste delicate flavours, I can only taste if something is salty has sugar in or if something is spicy… I couldn’t taste anything.’ (Emersyn)*



*‘I was eating, it was like there was nothing in my mouth, it was so bizarre. I didn’t notice my sense of smell, I didn’t realize how dependent your sense of taste is on your sense of smell. Because I could taste the saltiness or the sweetness for example, but within that no spices…’ (Meredith)*


For five participants, there was insufficient evidence to determine whether gustation was affected or unaffected, as they made no specific references to their perception of sweetness, saltiness, etc.

#### 3.1.3. Chemesthesis

There was good evidence that chemesthesis was unaffected, at least in respect of sensitivity to chili and pepper. Fourteen of the twenty participants commented on their intact sensitivity to chili, the heat of spices or other chemesthetic sensations, as demonstrated by the following examples.


*‘I know that chili causes a chemical reaction on your tongue so that doesn’t necessarily require smell to taste it, so I used to just pour chili and spicy stuff all over my food.’ (Emersyn)*



*‘…and my boyfriend sprayed his deodorant quite a lot and I could only tell when I could feel it in my throat too much…’ (Wendy)*



*‘…or like hot sauce because it was warm in my mouth.’ (Ella)*



*‘…my dad maybe cooked me a spicy meal one night and the spice was definitely still there.’ (Andre)*



*‘…spices made my mouth burn, so I’d have really spicy stuff just to experience something.’ (Meredith)*



*‘…like a can of coke, because that’s bubbly…’ (Alexadria)*



*‘…I opted for more spicy food… stuff with a lot more heat in it just to taste something.’ (Gena)*


There was insufficient evidence on chemesthesis in six of the narratives. There was no evidence pointing to significantly altered or absent chemesthesis.

### 3.2. Theme 2. Appetite (and Body Weight)

For effects on appetite, we took as evidence descriptions of appetite, motivation to eat and how much food was consumed, together with references to change or lack of change in body weight.

Some participants identified a loss of appetite that was independent of the altered taste. For example.


*‘When I first got the positive result back, I’d been feeling a bit off my food. So, I hadn’t been eating normally.’ (Emma)*



*‘On the Friday night I just felt really lethargic, and on the Saturday couldn’t literally get off the sofa.’ (Emma)*



*‘…it just really affected my appetite…I think because I was ill as well, but when you can’t taste, I just wasn’t hungry at all.’ (Martha)*



*‘I wasn’t really eating anything, so I couldn’t really tell if I’d lost my taste at the point…’ (Andre).*



*‘…but I didn’t want to eat anything anyway at the start, because my appetite had disappeared… I lost weight to begin with because I wasn’t eating, because I was ill anyway.’ (Meredith)*


Participants also identified decreased appetite due to altered taste perception. For example.


*‘I lost my appetite because I didn’t want to eat things that I enjoyed or drink things that I enjoyed, that didn’t have the taste that I was expecting. …I started losing quite a lot of weight.’ (Starla)*



*‘…feeling sick… was a really specific weird thing in relation to food…that’s really distressing when it tastes really disgusting.’ (Laurissa)*



*‘…literally I only ate when I got so physically hungry, that like I had to, there was no pleasure or enjoyment from it…I’d get irritated to have to eat, but obviously you need to eat to survive.’ (Ella)*



*‘I definitely ate less, and I didn’t really flavour anything because there’s sort of no point.’ (Melba)*



*‘I didn’t eat as much on the exact day I couldn’t taste just because it wasn’t enjoyable, it was just weird…’ (Kimberly)*



*‘…not being able to taste I wasn’t incentivized to eat anything that wasn’t put down in front of me.’ (Martha)*



*‘You kind of lose your appetite because if you can’t taste and smell things and you feel a bit weird, it’s all down hill for the food thing.’ (Norene)*


Other participants identified no change or an increase in appetite during anosmia. For example:


*‘I just kept things as normal, like I cooked the same food because of the family really. I’d just eat it…’ (Kallie)*



*‘…I actually found myself eating more because I was trying to find something that I could taste…healthy foods like fruit and vegetables stimulated nothing in taste so I would eat probably quite a lot of unhealthy foods…’ (Angela)*



*‘I definitely didn’t lose weight… I’d say I basically ate the same to begin with, and then I because it only lasted 7 days, the last few days I definitely got more bored of eating, but because I was bored and by myself I ate…’ (Marie)*



*‘…(during the period of anosmia) my appetite was unaffected, I was just as hungry as ever, so I would eat what I would usually eat.’ (Andre)*



*‘…when I lost my taste and smell I thought it was brilliant because I thought I was going to lose like five pounds over Christmas, but I was eating for the sake of eating and I couldn’t taste it at all.’ (Alexandria)*



*‘…when I was eating, I was eating more mindlessly I wasn’t thinking about it’ (Meredith)*



*‘…because I couldn’t taste anything it didn’t feel like I was eating anything as it was going down my throat, so I definitely think I had more to eat over that period of time… than what I usually eat.’ (Gena)*


Participants also commented on the appetising effect of the smell of food. For example:


*‘I really enjoy cooking and it just took all the fun out of it like you know when your smelling the ingredients as you’re making it, it builds up your appetite for the food. …Just meant it was so unexciting, didn’t enjoy cooking I thought it was such a chore.’ (Wendy)*



*‘…when you cook something and you smell it and it makes you hungry, that kind of thing has just only recently come back so when you’re cooking and you’re like oh my god that smells really nice.’ (Melba)*



*‘…he (Dianne’s husband) was doing like bacon sandwiches for us, but I couldn’t smell it, you know when the smell of bacon fills your house it makes you feel hungry, and I’d think I don’t really fancy it because I can’t smell it…’ (Dianne)*


### 3.3. Theme 3. Altered Food Choices

A theme related to appetite that emerged was altered food choices, driven by searching for (or in fewer instances avoiding) oro-sensory sensations from food, mainly in the form of texture, saltiness, sweetness and chemesthetic heat. For example:


*‘Definitely did change what I was eating, I was just eating really boring food and that mainly had interesting textures, not really interesting tastes more. I was eating more spicy food I think…’ (Wendy)*



*‘The texture, if you take avocado, without taste is horrible because it’s just like nothingness.’ (Laurissa)*



*‘I really tried to latch onto the salty and sweet things as much as I could.’ (Regana)*



*‘I remember going through like a jelly phase, just because jelly was like a texture to be eating.’ (Ella)*



*‘So it just meant that everything had to be about texture… I couldn’t eat like scrambled eggs or anything that was like really mushy… So I ate a lot of toast a lot of crisps just because of the crunch…’ (Angela)*



*‘I had like chicken and salad for lunch I could just taste the textures, so I got kind of bored of eating that kind of thing. …after a while I would only really eat things that were spicy or salty or sour things. So like popcorn for example, I could taste that was salty…’ (Marie)*



*‘I appreciated takeaways quite a lot, like food with stronger flavours sort of thing… like stronger salty flavours or like sweet flavours…’ (David)*



*‘I was putting loads of salt on my food because that was something that I could like taste a little…’ (Martha).*



*‘I wasn’t interested in eating very much but toast, maybe because the texture was maybe quite comforting, so something you’re used to… We were really into the texture.’ (Norene)*



*‘…having super sweet or super salty, like loads of soy sauce for example.’ (Meredith)*



*‘…I definitely went for much more spicy foods… like spicy stuff, curries, KFC because that’s got a lot of taste.’ (Gena)*


### 3.4. Theme 4. Drinks

#### 3.4.1. Alcoholic Drinks


*‘I stopped drinking wine, but I could taste white wine better so if I was going to have a meal, I would have white wine.’ (Emersyn)*



*‘…actually it’s funny about the Baileys (a flavoured alcoholic drink), because I did start drinking it after a few weeks and I couldn’t taste it properly and it was an almond soya one, and then I put it in the fridge and I tasted it after Christmas and I couldn’t believe that I was actually drinking (it), because it was when I could taste it better it was so disgusting.’ (Starla)*



*‘…it definitely made drinking alcohol easier but not nice. Like when you’re drinking a nice cocktail you want to be able to taste it, I couldn’t so it was a bit boring.’ (Wendy)*



*‘I could down a really strong vodka coke whatever, I could obviously feel the burn but there was just no flavour, which the burn’s ok it’s just the flavour that I hate… and wine just tasted like water which was great and so yes I ended up drinking a lot more…’ (Regana)*



*‘…it affected what alcohol I drank, like that was the time when I drank all the most horrible things we had round the house because nothing mattered again like what it tasted like.’ (Ella)*



*‘…I drank a lot of cheap wine because I couldn’t taste it…’ (Angela)*



*‘…white wine started to taste like really acidic to me and so I didn’t really enjoy drinking that.’ (Marie)*



*‘…so for example wine red or white…so you could sort of get hints of the flavour from drinking that but not much… but it definitely didn’t affect my drinking as such, if anything it made it worse.’ (Andre)*



*‘I had a glass of wine and I thought there’s no point kind of drinking it because I can’t taste it, but then I thought oh actually I can kind of taste it, but texture which was really weird.’ (Norene)*



*‘I didn’t enjoy drinking alcohol in the same way because I quite like choosing a nice beer, I just drank spirits at that point, so it did change what I was drinking.’ (Meredith)*


#### 3.4.2. Non-Alcoholic Drinks


*‘…it took the enjoyment out of everything. Even drinking a cup of coffee, like I wanted a cup of coffee but it just like didn’t taste like coffee it was just like a hot drink.’ (Kallie)*



*‘…I think it reduced the amount of like recreational drinks as in like fizzy drinks or juices I don’t remember having like any of that. I was purely drinking coffee for like the effects of caffeine.’ (Ella)*



*‘…I tried it again with more squash and when I didn’t taste it there was just no point, so I just stuck with water the whole day.’ (Kimberly)*



*‘Coffee does now taste different, yes coffee tastes really different, it sort of tastes a bit like bread kind of that sounds really weird doesn’t it that’s the closest thing I can kind of compare it to… Coffee is the one thing that has changed. …other than that, I don’t think I changed my drinking habits.’ (David)*



*‘…I didn’t drink any nice herbal teas like I normally would, it was like nothing drinking hot water.’ (Meredith)*



*‘I had no desire to drink anything, I didn’t feel thirsty… I didn’t drink as much, of anything.’ (Alexandria)*



*‘I have one coffee every morning and I can’t remember tasting it, but I put more sugar in it as well and again because I couldn’t taste it.’ (Gena)*


### 3.5. Theme 5. Eating Rituals

The theme of eating rituals comprised the following four sub-themes.

#### 3.5.1. Meal Times


*‘I’d missed meals, which was unusual for me…’ (Emma)*



*‘All of that went, there was no sitting at the table or doing anything…’ (Laurissa)*



*‘So yeah, it was really disruptive to my eating pattern and I think I could go a whole day without eating anything at all.’ (Alexandria)*



*‘I continued to eat normally because of my husband and kids, he was cooking, and we were still having breakfast, lunch and dinner so I was’ (Dianne)*


#### 3.5.2. Snacking


*‘…I stopped snacking…’ (Emersyn)*



*‘…after like supper normally, I really crave something really sugary like a piece of chocolate or like some fruit or something like that and I didn’t get that at all…’ (Martha)*



*‘I did continue snacking I think, just out of habit more than anything…’ (Dianne)*


#### 3.5.3. Social Bonding and Commensality


*‘I think it would have affected me socially, not appreciating food and things.’ (Kallie)*



*‘…when we eat we sit at the table and it’s a pleasant thing, it’s conversation where we chat…’ (Emma)*



*‘But there’s something about people eating together, it’s very sort of bonding… I think it really emphasized for me how much it’s social…’ (Laurissa)*


*‘…also it’s* (food) *such a social thing isn’t it, like such a talking point…’ (Angela)*


*‘…for certain events like a nice family BBQ which is like a nice wholesome thing in itself…’ (Andre)*


#### 3.5.4. Comfort Eating


*‘…before when I’d eat comfort food it was mainly about the taste, and now not really.’ (Emersyn)*


*‘Yes, I was still comfort eating, I just felt like it wasn’t fair, I was watching my husband eating them* (chocolates) *and I felt like if you can eat them then I’m not going to not eat them.’ (Kallie).*


*‘…when I was eating that risotto… it was comforting because risotto’s quite a comforting thing…’ (Laurissa).*



*‘…it made me more of a kind of comfort food eater…’ (Regana)*



*‘…I definitely didn’t get the same comfort from food.’ (Ella)*



*‘…I still did comfort eating, but I just couldn’t taste anything.’ (Angela)*



*‘…I would try and eat healthy, but I remember eating a lot more comfort food because I remember thinking you’re in so much pain when you eat you may as well eat something you really enjoy, and so I was having so many biscuits in one go.’ (Kimberley)*



*‘one big aspect of food’s comforting nature is gone, so yes it was quite depressing for a while.’ (David)*



*‘…my boredom and stuff when I had it… I think maybe it made it (comfort eating) worse, I don’t know as in I ate more, but also it doesn’t really give you that same like… if you can’t taste it…’ (Marie)*



*‘texture was maybe quite comforting, so something you’re used to, and something like that is quite comforting.’ (Norene)*



*‘there were times I was eating, and I couldn’t taste it, but I’d be stressed so I’d eat it anyway.’ (Meredith).*


### 3.6. Coping

Almost unanimously, participants coped well with their experience of altered chemosensory perception, in good part because they were confident that they would recover their sense of taste and smell in due course (within days or a few weeks). There was curiosity about the experience, and internet sources were mostly felt to be helpful. None of the participants sought professional help, although some tried ‘smell training’, which they had seen recommended. Support from partners and friends was appreciated.


*‘It didn’t impact me, and I didn’t get stressed about it all, as I said I was inquisitive and interested more than like horrified.’ (Emma)*



*‘I just hoped it would get better, I never took any nasal sprays, nothing like that I just kept reading up on it and others’ experiences to see what they were saying.’ (Kallie)*


*‘If it lasted anymore more than that* (10 days) *then I would have gone into fight mode and try and sort it out… But I definitely did a lot of googling, which I tried to not do because it does stress you out.’ (Wendy)*


*‘I didn’t get any help. We did try to test stuff like we kind of made a game out of it like we’d blindfold each other and see if you could guess what they were putting in your mouth…’ (Regana)*



*‘We basically googled it every day, because it was basically when it was becoming a symptom, just to see what you could do and how long it would last and actually quite a lot of my friends had it at the same time so we were all talking to each other about whether ours was getting worse or getting better.’ (Marie)*



*‘I looked up this like thing called smell therapy where you like smell different strong smells…’ (David)*



*‘…during the day when I’m not eating I wouldn’t think about it at all, so I didn’t really seek help or look up anything to try and solve it, I just kind of thought it will come back soon probably.’ (Andre)*



*‘I did loads of googling research and stuff and then I thought I just felt like the more stressed I got about it the less likely it would be to come back. So I just tried to forget about it…’ (Martha)*



*‘I did some research and there was stuff on smell training, so I got some essential oils like lemon and rose and I was doing that in the morning and the evening, but it felt really stupid because obviously I couldn’t smell anything…’ (Meredith)*



*‘It didn’t bother me too much, I mean I would get up every morning and smell that candle to see if I could smell that, but other than that I wasn’t too worried.’ (Dianne)*


## 4. Discussion

This study adds to the evidence on the effects of SARS-CoV-2 infection on human olfaction, gustation and chemesthesis [29], and importantly, it further characterises the impact of the loss and distortion of smell and taste perception on appetite and food choice. The effects on appetite, which were highly variable, provide insights into the role of smell, taste, chemesthesis and food texture in motivating food ingestion (and food rejection).

Our interviews confirmed that all participants had experienced altered olfaction. For all but one participant, this appeared to comprise a profound loss of olfactory perception (anosmia) for a period of time, rather than only diminished olfaction (hyposmia). This can be expected, given that we invited participants to join the study based on experience of ‘loss of taste and smell during COVID-19’. For one participant, their altered olfaction was dominated by parosmia and, later, included phantosmia. For a large majority of participants, the onset of anosmia was sudden (i.e., within at least a day), whereas the time course of recovery of olfaction was much more variable, being sudden for some participants and much more gradual (over months) for others, with only partial recovery for some individuals at the time of interview. Six participants reported perceptions consistent with parosmia and/or phantosmia. For four of these participants, the distortions of olfaction occurred only during recovery from anosmia.

As would be expected of people experiencing anosmia, all participants also reported altered taste perception. A common description was that food tasted ‘bland’, e.g., [30], or more extremely, that it tasted of ‘nothing’ or that it did not ‘taste of anything’. The extent to which altered gustatory perception additionally contributed to such perceptions is somewhat difficult to determine, however. On the one hand, such experiences might be explained solely by the loss of (retronasal) olfaction. As one of our colleagues remarked about their experience of eating a banana while anosmic with COVID-19: ‘it tasted of nothing… it was just a bit sweet’. Similar qualifying statements were made by several of the participants in this study. In other words, the experience of blandness, or even nothingness, may be due entirely to the profoundly reduced complexity of the flavour of foods that occurs when a person is anosmic. On the other hand, there was evidence indicating the presence of ageusia in three participants and the loss of bitter taste perception in one participant and the loss of sweet taste perception in another participant. For five participants, it was not possible to determine from the narratives whether gustation was altered.

Taken together, therefore, the present results indicate that olfaction is more vulnerable to SARS-CoV-2 infection than gustation is. This is consistent with conclusions from various studies, although our results suggest a lesser effect on gustatory perception (Table 2: 10 of 20 participants unaffected, with a lack of evidence for an effect for a further five participants) than most other studies, e.g., [3,4,5,18]. Survey studies may overestimate the extent to which gustation is affected for several reasons. This includes, for example, the general tendency to confuse the contribution of retronasal olfaction and gustation to the perception of flavour [17]. Relatedly, despite laying bare, for example, the sweetness of a food, the loss of olfactory function markedly reduces overall flavour intensity. It may be that this reduction in flavour is partly perceived as reduced sweetness. Furthermore, learned associations between olfactory and gustatory cues enhance, for example, the perception of sweetness. Caramel and vanilla have been shown to be particularly potent in this respect [31,32]. Anosmia would therefore be expected to reduce the perceived sweetness of foods containing such odourants. The mechanisms by which SARS-CoV-2 affects chemosensory perception remain to be fully elucidated [33].

Additionally, relevant to gustatory perception is evidence from the present study that at least some participants increased their intake of sweet and salty foods. This is consistent with results from other studies, e.g., [1,6,7,18]. Seeking out foods with high salt or sugar content, or increasing the amount of sugar and salt added to foods, could be interpreted as compensation for reduced perception of sweetness and saltiness. The narratives do not support this explanation, however. Rather, participants noted that they chose sweet, and more often, salty foods, to compensate for the lack of flavour they experienced from, for example, vegetables. References to sweetness and saltiness were far more common than references to bitterness and sourness, perhaps because the latter are less salient attributes of foods or are avoided. Therefore, we are unable to comment on the extent to which our participants experienced altered bitterness and sourness perception. There were a few references, however, indicating that savoury foods and foods containing monosodium glutamate were sought out, suggesting perception of umami was unaffected in those participants. Testing participants with COVID-19 with ‘pure’ (unimodal) gustatory stimuli of known concentrations is required to determine how commonly gustation is altered [2]. Similar testing for chemesthesis is also needed [2]. Clearly, such research would be difficult to conduct.

In the present study, 14 of the 20 participants made references to chemesthetic sensations, most often in describing the ‘heat’ or ‘burn’ of chili, pepper or ‘spices’. Indeed, like for sweetness, saltiness and umami, these references were often made in the context of adding ‘taste’ to foods. There was no specific evidence that chemesthetic function was diminished, but for six participants, the narratives contained no references relevant to chemesthesis, and overall there were few references made beyond those to the heat of chili, pepper or spices. It is possible that other aspects of chemesthetic perception were altered but went unnoticed. Selective chemesthetic loss would be consistent with the anatomy of chemesthesis [2,12].

In relation to assessing possible changes in appetite shown in Table 2, we took into account references to appetite, motivation and desire to eat, enjoyment of eating, and to the amount of food and/or calories eaten and changes in body weight. This is because food ingestion is motivated by more than the appreciation of the flavour of food in the mouth. In addition to the conscious experience of this ‘liking’, evidence points to a second, less obviously conscious, component of food reward, namely ‘wanting’ [34,35]. We define food reward as the momentary value of a food at the time of anticipated or actual consumption [36]. We found that liking (the pleasantness of the taste of a food) and hunger independently account for a large majority of the variance in food reward, with hunger (affecting wanting) being determined primarily by the amount of food present in the gut [36,37]. Anticipation of food reward is measured by ratings of desire to eat, and experienced food reward by ratings of meal enjoyment [38]. Importantly, food reward value is increased initially during consumption, depending on the detection of macronutrients arriving in the gastro-intestinal tract, followed by satiation, as food fills the upper gut. Rapid postingestively mediated increases in food reward are demonstrated by the phenomenon of ‘appetition’ [39] and by neuroanatomical research, e.g., [40].

Consistent with reduced food liking, a prominent theme for participants who reported a decrease in their appetite, and for others whose appetite was not reduced overall, was the loss of ‘enjoyment’ of eating, related to the blandness and, to a lesser extent, the distortion of taste (flavour) of familiar foods. This is unsurprising and has been observed in other studies of the effects of anosmia in participants with COVID-19, e.g., [1,7,18,41]. This was also expressed as lack of ‘excitement’ about eating by some participants. Furthermore, participants noted that their anticipation of eating (i.e., their anticipation of food reward) was dulled, which, for example, affected their enjoyment of food preparation. More directly, participants also commented that their hunger (desire to eat) was reduced because of the absence of the appetising effect of food smells.

By contrast, despite their anosmia, there was evidence that appetite was unaffected or increased for nine participants. Again, this is mostly consistent with findings from previous studies, e.g., [1,7,18,41]. There were two sub-themes clearly related to the maintenance or increase in food intake, both of which can be described as compensatory strategies. First, as noted above, there was the strategy of amplifying the oro-sensory qualities that were unaffected by anosmia, including sweetness, saltiness, chemesthetic heat and food texture. This included altered food choices, often described by participants as ‘unhealthy’, which comprised increasing intake of high sugar, high salt and highly ‘spiced’ foods. There were also descriptions of adding, in particular, salt, pepper and chili to foods. Additionally, increased importance was placed on food texture. For the participants in the present study, this involved seeking out ‘crunchy’, ‘crisp’ and ‘squidgy’ foods, but avoiding foods that, without flavour, were experienced as predominantly, for example, ‘slimy’ or ‘mushy’. These compensatory behaviours have been described in previous studies of acute COVID-19-associated chemosensory loss, e.g., [1,7,41].

The second sub-theme concerning compensation for the impoverished experience of food flavour resulting from anosmia was to increase the amount eaten. On the face of it, eating more of something that is experienced as less pleasant in the mouth (less liked) appears contradictory. Nonetheless, the idea of eating more in response to a diminished experience of food reward has been proposed as an explanation of obesity. On the other hand, this reward deficit theory of obesity is not well supported by current evidence [42]. Instead, consistent with other studies [1,41], our evidence of increased calorie intake may be at least partly explained by the strategy of choosing foods higher in salt and sugar, which tend to be energy rich, while reducing intake of ‘bland’, energy-dilute fruits and vegetables. Foods with high energy density are less satiating calorie-for-calorie than foods with low energy density, which may well make them more ‘wanted’, as well as more ‘liked’ [36,37]. The latter could compensate for the degraded experience of food flavour during acute anosmia, with the greater wanting further promoting intake. This wanting is maybe embodied in the experiences of eating ‘mindlessly’, eating ‘without thinking about it’ and ‘eating out of habit’, described by some of our participants. Additionally, relevant here is the effect of body weight changes on appetite, and in particular, increased appetite that occurs with body (fat) loss [43]. In our model of appetite and weight control, body weight loss results in increased food reward, which, all else being equal, will stimulate food intake, and thereby counteract the weight loss [36,37,44].

For two participants, decreased appetite preceded any obvious anosmia or other change in chemosensory perception. This is consistent with the decrease in appetite that can also occur, for example, during influenza. Such illness-related loss of appetite, or cachexia, is associated with inflammation and is likely mediated by certain cytokines acting on hypothalamic pathways [45]. In other words, this is a ‘top-down’ effect related to inflammation (which might also play a significant role in causing reduced appetite in older age [45]). Appetite recovered in these two participants after a few days, despite the onset of anosmia. It is possible that persisting decreases in appetite experienced by other participants were at least partly mediated by this mechanism, independent of anosmia.

Finally, disgust arising from the altered flavour of familiar foods is likely to have contributed to decreased appetite for some participants. The word ‘disgusting’ was used frequently and directly in reference to the changed flavour of foods. Food disgust, which has the function of protecting against ingestion of harmful foodstuffs, for example, spoiled or otherwise contaminated foods, is hard to overcome [46]. Information indicating potential contamination, even if contradicted by rational knowledge, is likely to lead to food rejection [46,47]. Thus, a familiar food, which does not taste as expected, will arouse disgust, perhaps especially so if this is not counteracted (compensated for) by the presence of innately liked sweetness or saltiness. There was evidence that parosmia and phantosmia [26] also contributed to feelings of disgust for some participants.

A further noteworthy theme that emerged from the narratives was participants’ altered relationships with alcoholic drinks. Some participants were clear that their alcohol intake had increased because reduced flavour intensity made high alcohol and/or lower-quality alcoholic drinks more acceptable (i.e., they tasted less unpleasant). In other words, the aftereffects of ingestion of alcohol dominated over flavour as a motive for consumption. On the other hand, a smaller number of participants reduced their intake of alcohol. They were much more likely to describe the flavour of, for example, wines, beers and cocktails as having become less subtle and less pleasant. For them, impoverished flavour affected enjoyment of the drink. Similarly, participants also described reduced enjoyment of and motivation to consume non-alcoholic drinks, including caffeine-containing drinks and, for example, fruit-flavoured ‘squash’ drinks, with some mentioning compensatory strategies, such as adding sugar or increasing the concentration (reducing the dilution) of these drinks. The narratives thus suggest that intake of non-alcoholic drinks was reduced overall, with one participant even noting that they ‘didn’t feel thirsty’. This is consistent with apparent thirst and generally higher than required fluid intake (i.e., fluid intake in excess of the requirement to maintain fluid homeostasis) being driven by the flavour, especially the sweetness, of widely consumed soft drinks [37].

We named the fifth and final theme we identified ‘eating rituals’. This included the sub-themes of mealtimes and snacking, which for some participants were altered seemingly because of reduced motivation to eat, whereas for others, the usual pattern of eating was maintained out of ‘habit’ and the continuation of family mealtimes. Linked with this was a sub-theme of ‘commensality’ (the act of eating with others), which demonstrated the value that participants placed on the sociability of mealtimes, together with the impression that this helped to maintain their motivation to eat, despite the impoverished experience of eating per se. The same was apparent for a fourth sub-theme, namely ‘comfort eating’. Relatively few descriptions of comfort eating mentioned negative emotional states; rather, they mostly referred to eating for pleasure and how this was degraded, and in some instances, compensated for by altered food choices. Again, it seems that reward, here articulated as ‘comfort’, was gained from eating, despite the taste (flavour) of food in the mouth evoking little or no pleasure.

Despite the varied and significant effects on eating and drinking, participants generally reported little concern about their altered taste and smell. They expected this not to be a long-lasting symptom. Their coping was facilitated by the social support they received from partners and friends, and by access to information shared on the internet.

## 5. Summary and Conclusions

In summary and conclusion, the present study confirmed the experience of all 20 participants of altered olfaction coincident with confirmed COVID-19. All but one participant experienced anosmia of variable duration, with evidence of ageusia or hypogeusia for five participants and no firm evidence of impaired chemesthesis. It is likely that some studies have overestimated the occurrence of altered gustatory and chemesthetic perception, at least in substantial part due to the tendency of respondents to confuse taste (gustation) and smell (retronasal olfaction). The onset of anosmia was mostly sudden, usually occurring after the appearance of other COVID-19 symptoms. Time to recovery of olfactory function was highly variable and accompanied by parosmia and/or phantosmia in a quarter of participants. Effects on appetite (food reward) were also variable, with participants experiencing decreased, no change or increased appetite. For two participants, decreased appetite preceded the onset of anosmia, which is consistent with evidence of an association between inflammation and diminished appetite. There was also evidence that appetite was affected by reduced enjoyment of food and drink due to participants’ anosmia. Consistent with findings from other studies, participants reported compensating for the diminished food flavour by choosing highly salty, highly sweet and ‘spicy’ foods, and increasing the amount of, for example, salt and chili added to foods. There was also increased attention to the texture of foods, with crispiness being highly favoured. In addition to these strategies for maintaining food liking (i.e., the pleasantness of the taste of food in our mouth), there was evidence that food ‘wanting’ was undiminished. This component of food reward is related to the postingestive detection of nutrients during a meal. We are less conscious of this mechanism than of liking, but phrases such ‘mindless eating’ were indicative of wanting. Some participants mentioned increased alcohol intake, in part facilitated by the reduced intensity of disliked flavours of alcoholic drinks. The narratives also underlined the value placed on ‘eating rituals’, including comfort eating, and the sociability and structure (routine and habit) associated with daily meals. Taken together, the results of the present study add to the record and analysis of lived experiences of altered chemosensory perception resulting from SARS-CoV-2 infection, which furthermore provide insights concerning the role of smell and flavour in motivating and rewarding food ingestion.

## Figures and Tables

**Table 1 foods-11-00607-t001:** The semi-structured interview questions, prompts and probes ^1^.

Questions	Prompts and Probes
How did you first become aware you were losing your sense of smell or taste?	
How long did each symptom last?	
Did it affect what you were eating?	
Did other sensory properties of food become more important?	For example, texture or temperature?
Did it affect the amount of food you ate?	Did you notice any subsequent change in your weight?
Did it affect your mood?	
Did it affect your eating rituals?	For example, cooking, mealtimes, food shopping?
Did it affect what you were drinking?	For example, tea, coffee or alcohol?
Did you notice the loss of smell day-to-day?	In other aspects of life, other than food?
Did it affect your appreciation of smell or taste?	
How did you cope with it?	Did you seek help?
Would you describe yourself as a comfort eater? Did it affect your quality of life? Is there anything else I should have asked you that I haven’t included?	Was this affected?

^1^ Actual conversations often deviated from the interview schedule. Questions, prompts and probes were flexible, depending on participant responses.

**Table 2 foods-11-00607-t002:** Summary of altered chemosensory perception and appetite associated with COVID-19 in study participants.

Participant ^1^	Olfaction	Gustation	Chemesthesis	Appetite	Notes
Emma (F, 49)	↓	→	→	↓	Decreased appetite preceded anosmia. Anosmia present for 2–3 weeks. Lost weight.
Emersyn (F, 21)	↓	↓ bitter	→	→	Anosmia an early symptom. No significant recovery of olfaction at time of interview after 4–5 months, with onset of parosmia after approximately 3 months.
Kallie (F, 40)	↓	→	→	→	Other symptoms preceded anosmia. Anosmia present for 2 weeks, accompanied initially by phantosmia? Gradual recovery of olfaction, accompanied by parosmia.
Starla (F, 48)	↓	↓	?	↓	Sudden onset of anosmia a few days to a week after onset of other symptoms. Partial recovery of olfaction after > 4 weeks. Lost weight.
Wendy (F, 21)	↓	→	→	→	Anosmia an early symptom. Anosmia present for 10 days.
Laurissa, (F 57)	↓	→	?	↓	Sore throat preceded sudden onset of parosmia by one day. Altered sense of taste linked with nausea and food disgust. Recovery of olfaction after 10 days, accompanied by phantosmia. Ate less, lost weight.
Regana, (F, 21)	↓	→	→	↑	Other symptoms preceded anosmia by one week. Onset and recovery from anosmia rapid, each within one day. Anosmia present for 10 days. Weight increased.
Ella (F, 21)	↓	?	→	↓	Other symptoms preceded anosmia. Anosmia present for 3 weeks. Weight decreased.
Angela (F, 21)	↓	→	→	↑	Sudden onset of anosmia 3 days after other symptoms. Anosmia present for at least several weeks. Gradual recovery of olfaction, accompanied by persisting parosmia and phantosmia. Weight increased.
Melba (F, 22)	↓	↓	?	↓	Anosmia an early symptom. Gradual recovery of olfaction over 3–4 months, accompanied by phantosmia still present at interview after 6–7 months. Small weight loss.
Marie (F, 21)	↓	→	→	→	Sudden onset of anosmia preceded other symptoms. Gradual recovery of olfaction. Weight unchanged.
Kimberly (F, 22)	↓	↓	?	↓	Sudden onset of anosmia and ageusia lasting 1 day. Preceded by other symptoms by 3 to 4 days. Sore throat made swallowing painful.
David (M, 21)	↓	→	→	↓	Sudden onset of anosmia half a day after onset of other symptoms. No symptoms after 2 weeks, except for anosmia. Partial recovery of olfaction, accompanied by parosmia, over 5–6 months. Weight decreased.
Andre (M, 22)	↓	?	→	↓ →	Anosmia/ageusia? present for 2 weeks. Preceded by other symptoms. Ate very little during first 4–5 days because felt ‘badly ill’.
Martha (F, 21)	↓	↓ sweet	?	↓	Sudden onset of anosmia, following onset of other symptoms. Gradual recovery of olfaction over 5 months, accompanied by parosmia.
Norene (F, 68)	↓	?	→	↓	Gradual onset of anosmia, beginning at same time as other symptoms. Gradual recovery of olfaction.
Meredith (F, 25)	↓	→	→	↓→	Loss of appetite preceded anosmia. Anosmia first apparent 5–6 days after onset of other symptoms. Anosmia present for 4 months, followed by full recovery of olfaction over several days.
Alexandria (F, 30)	↓	?	?	→	Other symptoms preceded onset of anosmia by a ‘few days’. Anosmia present for 4 days. Olfaction recovered before remission of other symptoms, which lasted for up to 2–3 weeks. Weight unchanged.
Dianne (F, 37)	↓	?	→	→	Sudden onset of anosmia 5 days after testing positive for COVID-19. Anosmia present for 5 days. Weight unchanged.
Gena (F, 35)	↓	→	→	↑	Anosmia present for one week.

^1^ Participant pseudonym, gender and age (years). ↑ Evidence for increase; ↓ Evidence for decrease; → Evidence for no change; ? Insufficient evidence. Further details are provided in the text.

## Data Availability

The data presented in this study are available on request from the corresponding author. The data have not been made publicly available to protect participant privacy.

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
