# Peer review of "Varied Effects of COVID-19 Chemosensory Loss and Distortion on Appetite: Implications for Understanding Motives for Eating and Drinking"

_foods, 2022, doi:10.3390/foods11040607_

Round 1

Reviewer 1 Report

Turner & Rogers: Varied effects of COVID-19 chemosensory loss on appetite: implications for understanding motives for eating and drinking

I enjoyed reading this well-written paper on chemosensory effects on appetite associated with COVID-19. Although the sample size was pretty small, I liked the study because of the details provided in the text and tables. I have only a couple of minor comments. 

Title: It was not always “chemosensory loss on appetite.” Perhaps the authors should improve the title a little bit. 

Line 37-81: These paragraphs represent a technically perfect story about many technical details of smell and taste. However, this is vaguely linked with the aim/-s of the paper. Adding a couple of concluding sentences at the end of each paragraph may help the reader better follow the main idea of the Introduction and the narrative of the whole paper. 

Lines 97-103: Remove these lines. 

Lines 628-653: These lines belong to the Results and must be rewritten to fit the style of Conclusions.

Author Response

Responses to Reviewer 1

Thank you for your positive helpful comments on our paper. We have made revisions in line with your recommendations, which are outlined in italics point by point below.

I enjoyed reading this well-written paper on chemosensory effects on appetite associated with COVID-19. Although the sample size was pretty small, I liked the study because of the details provided in the text and tables. I have only a couple of minor comments. 

Twenty individuals is generally regarded as a sufficient number of participants in qualitative research. We are confident that no new significant themes would have emerged with further interviews (e.g., lines 106-108 and 135-137). Our data comprised narratives totalling 31,000 words. We have now included this information on line 131.

Title: It was not always “chemosensory loss on appetite.” Perhaps the authors should improve the title a little bit. 

We have included ‘distortion in the title, to reflect the experience of parosmia and phantosima of some participants.

Line 37-81: These paragraphs represent a technically perfect story about many technical details of smell and taste. However, this is vaguely linked with the aim/-s of the paper. Adding a couple of concluding sentences at the end of each paragraph may help the reader better follow the main idea of the Introduction and the narrative of the whole paper. 

We have hopefully made the purpose of these paragraphs clearer, so that they better orient the reader as to what to expect, by starting the first of these paragraphs with this sentence ‘To set the scene for this research, the next three paragraphs describe relevant aspects of chemosensory (taste and smell) perception and its role in motivating and guiding eating behaviour.’ We think this (a clear understanding of taste and smell perception) is critical for interpreting data on chemosensory loss and distortion. Indeed, like Green for example, we imply that this has potentially misled interpretation of results of some studies on Covid-10. This is why we think this information needs to be placed at the start of the paper.

Lines 97-103: Remove these lines. 

Thank you! – we have done this.

Lines 628-653: These lines belong to the Results and must be rewritten to fit the style of Conclusions.

We think it is important, and helpful for the reader, to summarise the main findings along with our conclusions, especially given the variety of results that came out of the study. Accordingly, we have retitled the section Summary and conclusions.

Reviewer 2 Report

foods-1602485-peer-review-v1.comments

Varied effects of COVID-19 chemosensory loss on appetite: implications for understanding motives for eating and drinking

Lydia Turner, Peter J. Rogers *

Comment 1: Line No. 11: Study seems of great interest as little efforts have been given towards this scenario, as it has been mentioned that anosmia was considered. Why you have not judged the other changes associated with sense of smell, like parosmia and dysosmia? Because anosmia can do alter the intake but parosmia and dysosmia are more likely to be related with decreased intake.

Comment 2: Line No. 22: Please link your conclusive results on nutritional status outcomes of patients here under the point (contributes insights)

Comment 3: Line No. 29: Suggested to cite here the prevalence of major symptoms (The most dominant symptoms (>80%) were fatigue, fever and loss of senses) from the previous work of: Rabail, R., et. al. (2021). Nutritional and lifestyle changes required for minimizing the recovery period in home quarantined COVID-19 patients of Punjab, Pakistan. Food Science & Nutrition, 00, 1–24. https://doi.org/10.1002/fsn3.2458

Comment 4: Line No. 65: You should define the altered smell related terms (anosmia, parosmia, dysosmia etc.) once here as you are pointing out the changes in smell.

Comment 5: Line No. 90: Aren’t 20 participants very small sample size here? How can you justify the result outcomes on larger population?

Comment 6: Line No. 125: 9-25 minutes for interview time duration as you have mentioned here; It sounds not justifying as 9 minutes are quite shorter period of time to get satisfactory outcomes as compared to 25 minutes, even 25 minutes. It is suggested to crosscheck the 9 minutes and/or mention the mean value for time duration.

Comment 7: Line No. 127: Please add a brief summery on your questionnaire in your methodology or at least add relevant questions and number of total questions asked in the complete session.

Comment 8: Line No. 140:  Have you explained somewhere in your paper the outcomes of this question (How did you cope with it?). If not please do add a heading with answers and the treatment adapted or the foods used or the remedies that helped to increase intake and feel comfortable in altered sensations.

Comment 9: Line No. 309: The paper is conclusive and good in quality, should be modified as suggested.

Author Response

Responses to Reviewer 2

Thank you for your positive and helpful comments on our paper. We have made significant revisions in line with your recommendations, which are outlined in italics point by point below.

Comment 1: Line No. 11: Study seems of great interest as little efforts have been given towards this scenario, as it has been mentioned that anosmia was considered. Why you have not judged the other changes associated with sense of smell, like parosmia and dysosmia? Because anosmia can do alter the intake but parosmia and dysosmia are more likely to be related with decreased intake.

We do note the presence of parosmia and phantosmia, in various places in the manuscript, including defining these terms in the introduction. We have now also included the terms parosmia and phantosmia in the abstract.

Comment 2: Line No. 22: Please link your conclusive results on nutritional status outcomes of patients here under the point (contributes insights)

We provide information on weight changes in Table 2, and we have now included the summary in the abstract that 6 participants experienced weight loss. This is the only measure of nutritional status that we have. Our research was focused on experiences of a chemosensory loss and distortion, and effects on appetite, rather than nutritional status per se.

Comment 3: Line No. 29: Suggested to cite here the prevalence of major symptoms (The most dominant symptoms (>80%) were fatigue, fever and loss of senses) from the previous work of: Rabail, R., et. al. (2021). Nutritional and lifestyle changes required for minimizing the recovery period in home quarantined COVID-19 patients of Punjab, Pakistan. Food Science & Nutrition, 00, 1–24. https://doi.org/10.1002/fsn3.2458

The participants in this study were selected on the basis that they had experienced altered taste and smell perception. We did not systematically record other symptoms. Some of background papers we cite have information on the prevalence of other symptoms of COVID-19 in populations.

Comment 4: Line No. 65: You should define the altered smell related terms (anosmia, parosmia, dysosmia etc.) once here as you are pointing out the changes in smell.

We define these terms on lines 89-92 of the revised manuscript.

Comment 5: Line No. 90: Aren’t 20 participants very small sample size here? How can you justify the result outcomes on larger population?

This was a qualitative study, involving interviews rather than a questionnaire. Twenty individuals is generally regarded as a sufficient number of participants in qualitative research. Indeed, many studies interview fewer participants. Interviewing is stopped when no new themes emerge (saturation, lines 106-108 and 135-137). Our data comprised narratives totalling 31,000 words. We have now included this information on line 131.

Comment 6: Line No. 125: 9-25 minutes for interview time duration as you have mentioned here; It sounds not justifying as 9 minutes are quite shorter period of time to get satisfactory outcomes as compared to 25 minutes, even 25 minutes. It is suggested to crosscheck the 9 minutes and/or mention the mean value for time duration.

All 20 interviews generated sufficient information to contribute to the analyses. Even the shortest interviews approached 1000 words.

Comment 7: Line No. 127: Please add a brief summery on your questionnaire in your methodology or at least add relevant questions and number of total questions asked in the complete session.

The questions and prompts that helped structure the interview are listed in Table 1.

Comment 8: Line No. 140:  Have you explained somewhere in your paper the outcomes of this question (How did you cope with it?). If not please do add a heading with answers and the treatment adapted or the foods used or the remedies that helped to increase intake and feel comfortable in altered sensations.

Thank you for this suggestion. Although, strictly, it did not emerge as a theme in out thematic analysis, we have now included an analysis of responses to the question about coping. The results are on lines 438-468, with a short discussion on lines 652-655.

Comment 9: Line No. 309: The paper is conclusive and good in quality, should be modified as suggested.

Thank you.

Round 2

Reviewer 2 Report

Great work.